# Exercise Fat Oxidation Is Positively Associated with Body Fatness in Men with Obesity: Defying the Metabolic Flexibility Paradigm

**DOI:** 10.3390/ijerph18136945

**Published:** 2021-06-29

**Authors:** Isaac A. Chávez-Guevara, Rosa P. Hernández-Torres, Marina Trejo-Trejo, Everardo González-Rodríguez, Verónica Moreno-Brito, Abraham Wall-Medrano, Jorge A. Pérez-León, Arnulfo Ramos-Jiménez

**Affiliations:** 1Chemical Biological Sciences PhD Graduate Program, Department of Chemical Sciences, Biomedical Sciences Institute, Ciudad Juarez Autonomous University, Chihuahua 32310, Mexico; isaac.chavez@uacj.mx (I.A.C.-G.); awall@uacj.mx (A.W.-M.); alberto.perez@uacj.mx (J.A.P.-L.); 2Faculty of Physical Culture Sciences, Autonomous University of Chihuahua, Chihuahua 31000, Mexico; rhernant@uach.mx; 3Faculty of Sports, Autonomous University of Baja California, Mexicali, Baja California 21289, Mexico; marina.trejo@uabc.edu.mx; 4Faculty of Medicine and Biomedical Sciences, Autonomous University of Chihuahua, Circuito Universitario, Campus II, Chihuahua 31109, Mexico; evegonzal@uach.mx (E.G.-R.); vmoreno@uach.mx (V.M.-B.)

**Keywords:** energy metabolism, indirect calorimetry, physical exercise, antiobesity agents, aerobic exercise

## Abstract

Obesity is thought to be associated with a reduced capacity to increase fat oxidation in response to physical exercise; however, scientific evidence supporting this paradigm remains scarce. This study aimed to determine the interrelationship of different submaximal exercise metabolic flexibility (Metflex) markers and define its association with body fatness on subjects with obesity. Twenty-one male subjects with obesity performed a graded-intensity exercise protocol (Test 1) during which cardiorespiratory fitness (CRF), maximal fat oxidation (MFO) and its corresponding exercise intensity (FATmax) were recorded. A week afterward, each subject performed a 60-min walk (treadmill) at FATmax (Test 2), and the resulting fat oxidation area under the curve (TFO) and maximum respiratory exchange ratio (RER_peak_) were recorded. Blood lactate (LA_b_) levels was measured during both exercise protocols. Linear regression analysis was used to study the interrelationship of exercise Metflex markers. Pearson’s correlation was used to evaluate all possible linear relationships between Metflex and anthropometric measurement, controlling for CRF). The MFO explained 38% and 46% of RER_peak_ and TFO’s associated variance (*p* < 0.01) while TFO and RER_peak_ were inversely related (*R*^2^ = 0.54, *p* < 0.01). Body fatness positively correlated with MFO (*r* = 0.64, *p* < 0.01) and TFO (*r* = 0.63, *p* < 0.01) but inversely related with RER_peak_ (*r* = −0.67, *p* < 0.01). This study shows that MFO and RER_peak_ are valid indicators of TFO during steady-state exercise at FATmax. The fat oxidation capacity is directly associated with body fatness in males with obesity.

## 1. Introduction

Pandemic obesity is a premorbid condition linked to a wide range of metabolic disturbances and premature death [1]. An impaired metabolic flexibility (Metflex) to channel energetic substrates (fat/carbohydrate) according to energy expenditure and macronutrient metabolic availability has been associated with obesity [2]. Conversely, a reduced fat oxidation capacity (a.k.a. metabolic inflexibility (Metinflex)) is a typical feature in obesity due to a reduced muscle mitochondrial volume-density, adipogenesis, and reduced activity of carnitine palmitoyltransferase-1 (CPT-1), citrate synthase, and cytochrome c oxidase [3,4]. Metinflex may promotes the deposition of ceramides and diacylglycerol in skeletal muscle resulting on insulin resistance [5], although systematic evidence contradicts the aforementioned paradigm.

The fat oxidation increase in response to submaximal exercise intensity is presumably blunted in obesity; however, Arad et al. [6] performed a systematic review on this matter observing that just one-third of the included case (obese)–control (lean) studies stand for Metinflex in subjects with obesity. The authors commented that (i) the methodological approaches (e.g., different exercise intensities and ergometer type used), (ii) the employed variables to assess Metflex [e.g., maximal fat oxidation (MFO), total fat oxidation (TFO), and respiratory exchange ratio (RER)], (iii) the participants’ characteristics (e.g., race, sex, and body composition), and (iv) differences in cardiorespiratory fitness (CRF) of the subjects could explain the discrepancies among studies. With respect to methodological approaches, different Metflex markers have been defined from the fat oxidation and RER kinetics measured during submaximal intensity. Particularly, MFO and its corresponding exercise intensity (FATmax) (Figure 1A) assessed through a short-stages incremental-load test (1–10 min) [7] have been proposed as valid Metflex markers [8], although the relationship between MFO and TFO (Figure 1B) during a steady-state exercise session has not been studied. Indeed, because fat oxidation exponentially decreases below MFO after 10 min of continuous FATmax exercise, (Figure 1B) [9,10], the reliability of MFO to represent fat oxidation capacity has been questioned.

Moreover, whether MFO and TFO are related to dynamic substrate switching (stage 1 (onset of exercise) and 2 (time-dependent fat oxidation increase)) for energy production during steady-state exercising remains unexplored [11,12]. The time lapsed (latency) until the maximum RER value (RER_peak_) and the subsequent fat oxidation increase have been recently proposed as Metflex markers (Figure 1C), suggesting that metabolically flexible individuals will exhibit a faster switch towards fatty acid metabolism and a large fat oxidation increment within a specific time interval [13,14]. Nonetheless, controlled trials in our laboratory revealed that some individuals exhibit an oscillatory RER behavior without a significant RER increase at the onset of exercise (Figure 1D). Therefore, latency and fat oxidation increase cannot be objectively measured for all individuals, limiting its utility for Metflex assessment. Here, we suggest that maximum RER value (RER_peak_) observed at the onset of exercise could be a valid Metflex marker with a lower value indicating a higher fat utilization during steady-state exercise.

The existence of many Metflex markers hinders comparability among studies results. Thereby, it is necessary to elucidate, whether these Metflex markers are associated and truly represent the capacity to use fat as a fuel during exercise. From the above-mentioned markers, only MFO has been directly associated with body fatness, without observing a significant relationship among other variables [15,16,17]. In these studies, CRF has not been considered as a covariate, which is associated with fat oxidation [16,17], and with metabolic syndrome in males and females [18].

For the above-mentioned reasons, the present study aimed to determine the interrelationship among the MFO, TFO, and RER_peak_, and its association with physical fitness in men with obesity. We hypothesized that Metflex markers would be correlated and directly related with body fatness, because systematic evidence failed to demonstrate a low exercise fat oxidation in subjects with obesity.

## 2. Materials and Methods

### 2.1. Participants

Twenty-one Mexican males with obesity were recruited via institutional emails and websites, as well as through pamphlets and posters. Inclusion criteria were: (I) body mass index (BMI) ≥ 30 kg∙m^2^; (II) body fat percentage ≥ 30%; (III) fat mass index ≥6 kg∙m^−2^; (IV) not engaged in regular exercise training (<150 min per week). Participants reported their physical activity habits by recording their 24 h physical activities (i.e., transportation, scholar activities, labor duties, and domestic tasks) during seven consecutive days (Appendix A). Moreover, the participant’s health status was assessed through a medical history (Appendix A) and none of them reported a clinical background of cardiovascular, metabolic, or respiratory diseases. The general details of the study were provided to all participants, who signed a written informed consent after acceptance. Previously, the study protocol was approved by the Ethics Committee of the Autonomous University of Juarez (CIBE-2018-1-11) and all experimental procedures were carried out in accordance with the Declaration of Helsinki and the Mexican law for studies in human (NOM-012-SSA3-2012).

### 2.2. Study Design

Selected participants attended the laboratory on three occasions (Figure 2). Session 1: Participants underwent resting metabolic rate evaluation. Session 2: Subjects performed a graded-intensity exercise protocol during which CF, MFO, and FATmax were recorded. Session 3: Subjects walked at FATmax for 60 min on a treadmill for the assessment of macronutrient oxidation, blood LA_b_, oxygen consumption (VO_2_), and heart rate (HR) kinetics (see below). The second and third laboratory visits were separated by seven days while all the experimental sessions were done between 8:00 and 11:00 h after 10–12 h overnight fasting and the laboratory environmental temperature was kept between 22–24 °C.

Patients were instructed to maintain their habitual diet and refrain from any kind of physical exercise activities during their participation. Moreover, energy drink consumption was restricted a week before trials. Each subject consumed a control dinner with a balanced macronutrient content (55% carbohydrates, 30% lipids, and 15% proteins) overnight before both exercise tests. The meals were designed by a nutritionist and prescribed to provide the 30% of the total energy requirement (969 ± 148 kcal).

### 2.3. Anthropometric Measurements

Body weight and height were measured with a digital scale (SECA 876, Hamburg, Germany) and a wall mount stadiometer (SECA BM206, Hamburg, Germany). Body mass index (BMI; kg∙m^−2^) and fat max index were calculated (FMI; kg of body fat∙m^−2^). Fat mass and free fat mass (FFM) were assessed by air displacement plethysmography (BODPOD; Cosmed, Rome, Italy) in accordance with manufacturer guidelines. Percentage of body fat (%BF) was calculated using Siri’s equation [19].

### 2.4. Metabolic Oxidative Measurements

Gas exchange at rest and during both exercise tests was measured using indirect calorimetry with a gas analyzer (Cortex, MetaLyzer 3B, Germany). The system was calibrated before each test using certified gas mixtures of known concentrations (5% CO_2_, 16% O_2_, and balanced of N_2_; Cortex-Medical). A 3-L syringe (Hans Rudolph, Shawnee, USA) was used to calibrate the flow sensor. Oxygen uptake and carbon dioxide production (VCO_2_) were employed to calculate metabolic rate with the Weir equation [20], while macronutrient oxidation was calculated with Frayn’s stoichiometric equations [21] assuming that urinary excretion was negligible:-Fat oxidation:
$$\left(1.67\times VO_{2}\right)-\left(1.67\times VCO_{2}\right)$$

-Carbohydrate oxidation:

$$\left(4.55\times VCO_{2}\right)-\left(3.21\times VO_{2}\right)$$

#### 2.4.1. Resting Metabolic Rate Procedure

Participants remained sitting on a reclining chair for 10 min before metabolic measurements were taken for 15–20 min. The room lights remained off, and instrumental soft jazz was played during the entire measurement. Five continuous minutes of RER measurements with a CV ≤ 5% were considered as steady state for oxidative metabolic calculations.

#### 2.4.2. Incremental Exercise Test Procedure

The test was performed on a Quinton treadmill (TM55, Washington D.C., USA) to evaluate MFO, FATmax and peak of oxygen uptake (VO_2peak_). Prior to the incremental exercise test, a brief walking warmup (5 min) was carried out at 4 km∙h^−1^ with no inclination. The test started at 3 km∙h^−1^ at 1% gradient; afterward, the velocity was increased every 3 min by 1 km∙h^−1^ until 1.0 of RER was sustained for 30 or more seconds. Thereafter, velocity (by 1 km∙h^−1^) and gradient (by 1%) was increased simultaneously every 3 min until exhaustion, following the American College of Sports Medicine general indications for stopping an exercise test [22]. VO_2_, VCO_2_, HR (Polar Electro F6, Kempele, Finland), LA_b_, and perceived exertion by 0–10 Borg scale [23] were recorded at rest and during the entire test.

The VO_2peak_ was defined by averaging the VO_2_ values over the last 30 s of the test and the cardiorespiratory fitness level of the subjects was defined according to ACSM criteria [22]. The VO_2_ and VCO_2_ values registered on the last 120 s of each stage during this test (RER with CV≤ 5%) were used to calculate MFO (mg∙kg FFM∙min^−1^) and FATmax (Figure 1A), plotting fat oxidation against the relative exercise intensity (%VO_2peak_) [7]. VO_2_ at FATmax, and HR were used to control the exercise intensity during the 60-min FATmax trial (see below).

### 2.5. 60 min Treadmill Walking at FATMax

To standardize the initial metabolic conditions of the participants, a new resting metabolic rate and HR were measured before this trial. Then, a 5 min warmup was done at 4 km∙h^−1^ at a gradient of 0%. Afterward, patients exercised at FATmax for 60 min on the treadmill, sustaining their intensity (VO_2_ and HR) at corresponding FATmax (HR ± 5 beats∙min^−1^). Gas exchange was measured during the entire test for the assessment of energy expenditure and nutrient oxidation. The exercise session was divided into 5 min intervals. As above, the VO_2_ and VCO_2_ were recorded during the last 120 s of each interval (RER with CV ≤ 5%). The TFO was defined as 60-min fat oxidation area under the curve (FO_AUC_, Figure 1B). RER_peak_ was defined as the maximal RER value during the first 30 min of FATmax exercise, independently of the RER kinetics (Appendix A).

### 2.6. Blood Lactate Assay

Capillary blood samples were taken from the fingerprint before exercise initiation, every 3 min during the graded exercise test, and every 30 min during prolonged FATmax exercise. The LA_b_ was determined by using test strips and lactate plus meter (Nova Biomedical, Waltham, MA, USA). The latter was precalibrated with control lactate solutions (1.0–1.6 and 4.0–5.4 mM).

### 2.7. Statistical Analysis

The Shapiro–Wilk test, Q–Q, and box plots were used to analyze data distribution. The association of Metflex markers with body fat was explored through partial correlations analysis, controlling for VO_2peak_. Linear regression analysis was used to study the independent interrelationship of exercise Metflex markers. Repeated measures analysis of variance (ANOVA) with Bonferroni corrections was used to investigate the dynamic macronutrient oxidation during the 60 min of FATmax exercise. The 60 min of fat and carbohydrate oxidation area under the curve was computed with GraphPad Prism v. 8.1 (Harvey Motulsky, San Diego, CA, USA), and the rest of the analyses were performed with SPSS v. 22 (IBM corporation, NY, USA). The statistical significance was accepted at *p* ≤ 0.05. Figures were constructed in GraphPad Prism v. 8.1. Data were reported as mean ± SE (graphs), ± 95% CI (Tables), and ± SD (text).

## 3. Results

### 3.1. Participants Physical Fitness and FATMax

The physical fitness of the participants is shown in Table 1. All participants were considered obese (fat mass: 35.4 ± 7.0; fat mass index: 11.76 ± 3.62) and exhibited a low CRF (VO_2peak_: 39.9 ± 6.0 mL∙kg^−1^∙min^−1^) [22]. The MFO during the incremental exercise test was 4.14 ± 1.24 mg∙kgFFM^−1^∙min^−1^, and FATmax occurred at a low exercise intensity (35.85 ± 6.09% of *VO_2peak_* and 102 ± 11 beats∙min^−1^).

### 3.2. Metflex Markers

The MFO explained 38 and 46% of RER_peak_ (Figure 3A) and TFO’s associated variance (Figure 3B), while TFO and RER_peak_ were inversely related (Figure 3C). Body fatness positively correlate with MFO and TFO but was inversely related to RER_peak_. The FATmax was not related with any Metflex marker or body fatness (Table 2).

### 3.3. Substrate Oxidation at FATmax (Test 2)

According to VO_2_ and HR kinetics (Figure 4A,B), exercise intensity remained constant throughout Test 2 (CV ≤ 5%). Fat oxidation exponentially increased in the first five minutes, decreased (~36%) between the 10–20 min and subsequently increased until reaching FATmax at ~60 min (Figure 4C). Conversely, carbohydrate oxidation increased during the first 15 min but decreased afterward (Figure 4D). LA_b_ also increases slightly (~10%) above baseline concentrations and remains higher than FATmax throughout the test (Figure 4E). Total energy expenditure during Test 2 was 264.91 ± 66.89 kcal∙h^−1^ (95% CI: 234.46−295.35).

## 4. Discussion

### 4.1. Correlation of Metflex Markers

Different Metflex markers have been physiologically related to physical fitness and insulin sensitivity [2,4,6,8,10,13,14,24]. The present study aimed to determine their correlation particularly testing their association with body fatness while controlling for VO_2peak_ (mL∙kg^−1^∙min^−1^) in subjects with obesity. In this sense, Takagi et al. [9] and Ozdemir et al. [10] suggest that MFO and TFO do not correlate because the MFO is not sustained beyond 10 min of prolonged exercising at FATmax. On the contrary, this work demonstrates that 46% of TFO’s associated variance is explained by MFO, and the latter shows an inverse relationship with the RER peak (*p* < 0.01) and so, MFO is a valid marker of dynamic substrate selection (fueling) during submaximal exercise. Measurement of the fat oxidation increase in response to submaximal exercise is a valid strategy for Metflex assessment [2,4], future studies need to analyze whether MFO, TFO and RER_peak_ are associated with postprandial fat oxidation after a high fat meal and nutrient oxidation during the sleep period, conditions that require Metflex.

The present study also demonstrates that RER_peak_ at the onset of exercise is a valuable Metflex marker and subsequent studies may consider it for analyzing Metflex determinants and its associations with metabolic syndrome. Previous investigations proposed to use the time interval to RER_peak_ and the subsequent fat oxidation increase (Figure 1C) [13,14]. Nonetheless, in the present study, a hyperbolic pattern during steady-state exercise was not observed in all individuals (Appendix A). Indeed, in a complementary analysis, neither latency time nor fat oxidation increase were associated with MFO or TFO (Appendix A). Thus, future studies should consider RER_peak_ instead of latency and fat oxidation increase when assessing Metflex.

### 4.2. Obesity and Metflex

A decreased capacity to increase fat oxidation when fatty acid availability or energy expenditure increases has been associated with obesity and lipid storage in skeletal muscle that impairs insulin signaling [2,3,4,5]. In contradiction with this paradigm, we observed a positive association between body fatness and the fat oxidation increase in response to physical exercise at FATmax (Table 2). The results of the present study agree with Emerenziani et al., [25] who observed a higher MFO in women with class III obesity compared with class I and II individuals, and supports previous studies reporting a higher MFO [26] and TFO [27,28] in subjects with obesity when compared against lean counterparts, as well as a lower RER_peak_ during moderate-intensity exercise [27]. Rationale for a positive association between body fatness and exercise fat oxidation are: (I) increased free fatty acid circulating levels and (II) augmented intramuscular triglyceride oxidation [6,27,28]. Nevertheless, mechanisms explaining a higher intramuscular lipid oxidation in subjects with obesity requires further clarification.

On the other hand, the results of the present study differ with previous investigations performed on recreationally active overweight and lean subjects where MFO was not associated with body fatness [15,16,17]. Differences in the participants’ sex, body composition, physical activity level and experimental design, hinders comparison among studies result. Furthermore, Blaize et al., [15] and Croci et al., [17] included data from lean and overweight subjects in the correlation analysis while only subjects with obesity where included in the present study. Moreover, these studies did not include VO_2max/peak_ as a covariate in their analysis which may bias their results. Moreover, whether exercise fat oxidation is or not associated with body fat percentage may depend on the participants’ ethnicity [6], body fat distribution (i.e., high/low abdominal to lower body ratio) [29] and metabolic phenotype (i.e., low or high resting respiratory quotient [30].

Interestingly, before a reduced-fat oxidation capacity was associated with obesity, Schutz et al., [31] observed a positive correlation between fat mass and postprandial fat oxidation, suggesting that fat oxidation increases with obesity as a protective mechanism that allows the progressive attenuation of the impact of an excess fat intake on fat stores. The majority of studies evaluating whether Metinflex is associated with obesity, have used a cross-sectional design limiting our comprehension about whether Metinflex develops with fat mass increase leading to insulin resistance. In this sense, a recent study observed that eight weeks of over-feeding (40% above baseline energy requirements) increased the fat mass by 55%, which was accompanied by an augment in fat oxidation without affecting the Metflex; despite observing a significant decrease in insulin sensitivity [32]. The authors suggested that fat mass increase does not affect Metinflex, neither Metinflex was responsible for insulin resistance which defies the existing Metflex paradigm. Our data reinforce this idea as the ability to increase fat oxidation in response to physical exercise which augment energy expenditure was directly associated with body fat percentage. Whereby, insulin resistance resulting from lipid accumulation may be explained by other mechanisms.

The carbohydrate–insulin model of obesity [33] is a good theory that can explain increased skeletal muscle lipid accumulation. A high carbohydrate diet consumption leads to postprandial hyperinsulinemia that reduces long-chain fatty acids oxidation in skeletal muscle by inhibition of carnitine palmitoyl-transferase 1 due to increased malonyl-CoA, a precursor for de novo lipogenesis [34]. Further, as insulin promotes energy deposition, substrate disposal is rapidly reduced, leading to frequent hunger sensation and overfeeding [33], contributing to sustained inhibition of fatty acid oxidation. In fact, whether Metinflex to carbohydrate feeding can lead to ectopic lipid accumulation and insulin resistance is unclear [4].

Alternatively, the “unfolded protein response”, also known as endoplasmic reticulum stress response, may lead to an increase of lipids in discrete cell compartments, activating protein kinase θ, which would inhibit the Insulin Receptor Substrate and thus the canonical Insulin signaling pathway. It is tempting to think that the endoplasmic reticulum stress response is enhanced in males with obesity due to the higher availability of muscle lipids [35].

### 4.3. Kinetics of Macronutrient Oxidation during FATmax

This study observed two different kinetics on fat oxidation during exercise at FATmax: (I) fat oxidation rose to the maximal values during the first five minutes of physical exercise, then it decreased; (II) after 15–20 min the fat oxidation increased again, until reaching the MFO around 60 min of exercise onset (Figure 4C). Our data contradict the results from Ozdemir et al., [10], who studied sedentary lean subjects (%body fat ~18.7) and observed an exponential fat oxidation decay after five minutes of FATmax exercise without a subsequent fat oxidation recovery. Apparently, these differences suggest a higher fat oxidation capacity in subjects with obesity, nonetheless, the discrepancies might be explained by a higher participants’ CRF observed in the present study (~39.85 vs. 33.53 mL∙kg^−1^∙min^−1^) as FATmax was similar in both investigations (~102 vs. 103 beats∙min^−1^). Further research on fat oxidation kinetics in people with different fat mass degrees but controlling the CRF is needed.

At the onset of exercise, carbohydrate and fat oxidation simultaneously increase because metabolic rate augments several-fold above resting values [36]. This metabolic switch—called the “gross control phase” of aerobic exercise—is explained by the increase of sarcoplasmic Ca^2+^ prior to muscle contraction initiates, which upregulates glucose and fatty acid uptake and oxidation in muscle fibers [11]. Thereafter, a decrease in fat oxidation and augmented carbohydrate utilization is regularly observed [13,27,28]. The reasons are not well understood, nevertheless, the increase in glycolytic flux at exercise initiation may result in negative feedback inhibition of β-oxidation due increased mitochondrial FADH_2_, NADH, and acetyl CoA levels [37]. Further, an increase in glycolytic flux can decrease muscle pH which may reduce the activity of carnitine palmitoyl transferase-1 and citrate synthase [36]. Indeed, an increase in glycolytic flux, indicated by the augment in blood lactate during the first minutes after exercise onset, coincides with RER_peak_ [13,27,28]. A rapid desensitization and sequestration of the beta-adrenergic receptor have also been reported, due to an increase in its phosphorylation by β-adrenergic receptor kinase and cAMP-dependent protein kinase [38]. The subsequent and gradual increase in lipid oxidation may be due to the joint participation of increased adrenaline, growth hormone, glucagon, and intracellular calcium, to increase lipolysis in adipose tissue and skeletal muscle, and muscle beta oxidation [39,40,41].

The inverse association between body fatness and RERpeak may represent a lower glycolytic flux increase at the beginning of the exercise. Possible reasons are: (I) a lower skeletal muscle glucose uptake, and (II) a lower skeletal muscle glycogen utilization as observed by Goodpaster et al. [27]. This supports the idea of higher reliance on lipid metabolism for ATP production in obesity proposed by Schutz et al. [31]; nonetheless, future studies need to corroborate the physiological and molecular mechanisms that explain a lower RERpeak.

### 4.4. Study Strengths

It is well documented that fasting fat oxidation is sensitive to the macronutrient content of the preceding meals [4] and is positively associated with the fat utilization at FATmax [16]. Thus, an appropriate assessment of Metflex during exercise requires a prestudy diet standardization. For the present work, participants consumed a standardized meal the night before both exercise trials which resulted on similar resting fat oxidation (0.07 ± 0.02 vs. 0.06 ± 0.03 g∙min^−1^, *p* > 0.05) and resting energy expenditure (436.81 ± 59.87 vs. 464.18 ± 117.72 kj∙h^−1^, *p* > 0.05) among exercise sessions. Further, exercise intensity remained constant through the entire exercise tests (Figure 4A,B).

### 4.5. Study Limitations

As previously mentioned, the present study is limited to analyzing the association among the different Metflex indicators, nevertheless, it cannot define the biochemical and molecular mechanisms that explain such associations. Further studies need to analyze whether body fat percentage is associated with circulating free fatty acids levels, muscular fatty acid uptake, mitochondrial fatty acid transport and intramuscular triglyceride hydrolysis and oxidation. In addition, future studies should determine whether Metflex and body fat percentage are related with the genotype of the ADRB3, CD36 and ACE genes and the muscle content of CPT and ATGL which have been associated with MFO [42,43,44,45].

In addition, the here reported data is limited to young-adults’ males with obesity, further studies should corroborate the presented associations in other populations like females and adolescents, due the potential effect of ovarian hormones (17β-estradiol) on lipid metabolism [11,46] and the observed MFO reduction across the lifespan [47]. Moreover, future studies must analyze the here observed associations in subjects with metabolic syndrome and trained individuals. In addition, participants exercised on a treadmill for the assessment of Metflex and it has been demonstrated that energetic fuel metabolism differ among exercise modalities [48,49], partly because the adrenergic response to exercise depends on the amount of muscle mass recruited [50]. Therefore, future studies should include all the here analyzed variables during another exercise conditions like rowing or cycling.

On the other hand, the sample size of the present study is similar to previous studies analyzing the association of MFO and body fatness (sample size range 14–54) [15,16,17], nonetheless, future large-cohort studies are needed to corroborate the here reported associations.

## 5. Conclusions

The MFO assessed through a short incremental exercise test is a reliable indicator of TFO during prolonged FATmax exercise. These Metflex markers are inversely related with carbohydrate increase oxidation at the onset of exercise, indicating a higher reliance on lipid metabolism for ATP production. Thereby, RERpeak may be considered as a Metflex marker in future studies. Fat oxidation increase in response to physical exercise is positively related to body fatness in men with obesity. Therefore, the fat oxidation capacity does not decrease with fat mass expansion, and other mechanisms must trigger skeletal muscle lipid accumulation that leads to insulin resistance.

## Figures and Tables

**Figure 1 ijerph-18-06945-f001:**
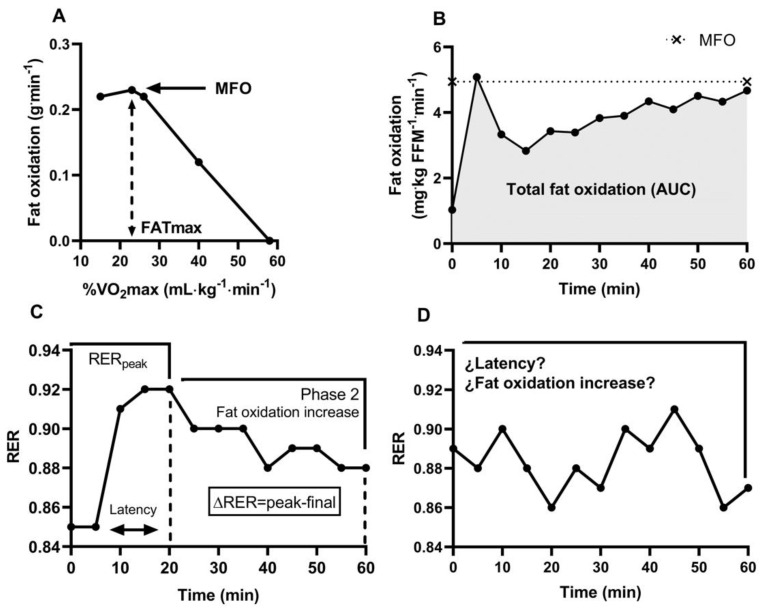
Assessment of metabolic flexibility during submaximal aerobic exercise. Footnotes: (**A**) Fat oxidation kinetics (incremental-load exercise test) vs. exercise intensity to identify maximal fat oxidation (MFO) and corresponding exercise intensity (FATmax). (**B**) Fat oxidation kinetics (AUC; steady-state exercise trial) to estimate total fat oxidation (TFO). (**C**,**D**) Respiratory exchange kinetics (RER; steady-state exercise trial). “Latency” is defined as the time interval used to reach RER_peak_ by individuals showing a hyperbolic pattern. Fat oxidation (as ΔRER) = RER_Net_ − RER_peak_. Author’s experimental data from incremental/constant FATmax tests with enrolled participants (see below).

**Figure 2 ijerph-18-06945-f002:**
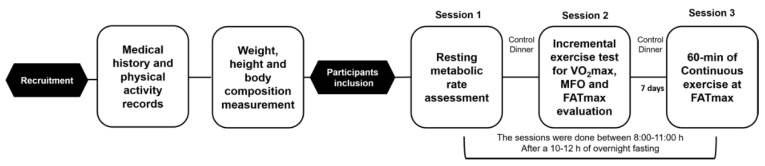
Stepwise experimental design.

**Figure 3 ijerph-18-06945-f003:**
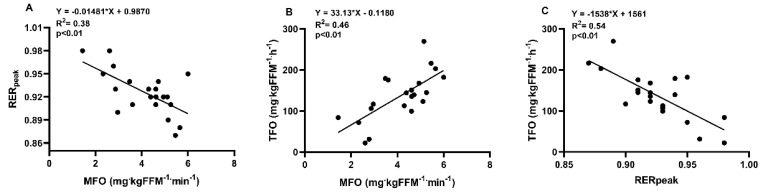
Linear regression between Metflex indicators. Maximal fat oxidation (MFO), total fat oxidation during 60 min (TFO), and maximum respiratory exchange ratio at the onset of exercise (RER_peak_).

**Figure 4 ijerph-18-06945-f004:**
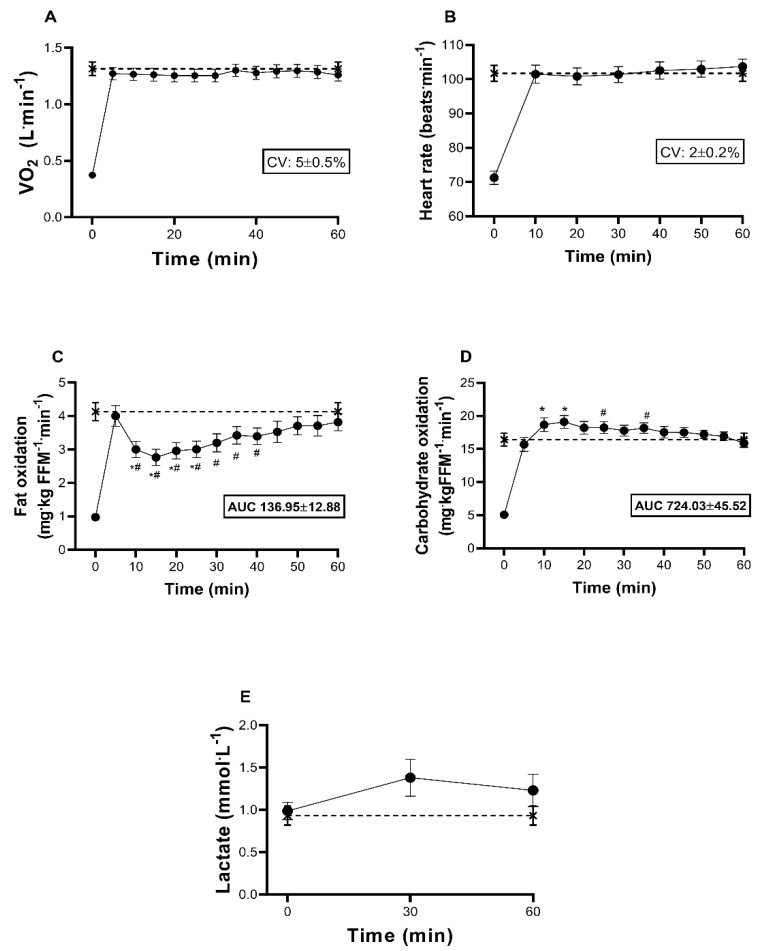
Time-trend changes in cardiorespiratory and substrate-using indicators at FATmax. 5 min interval during 1 h and each value is presented as mean ± SE (* *p* < 0.05 (vs. 5 min value); ^#^
*p* < 0.05 (vs. 60 min value)): oxygen uptake (**A**), heart rate (**B**), fat (**C**) and carbohydrate (**D**) oxidation and lactate production (**E**). Constant (solid line) and incremental-load (stripped line) exercise tests. Within-subject coefficient of variation (CV; **A**,**B**) and area under the curve (AUC; **C**,**D**).

**Table 1 ijerph-18-06945-t001:** Characteristics and physical fitness of enrolled participants.

Variable	Mean (95% CI)
Age (years)	27 (24−30)
Height (m)	1.75 (1.72−1.79)
Body mass (kg)	100.3 (93.8−106.7)
Body mass index (BMI; kg∙m^−2^)	32.6 (30.6−35.2)
Fat mass (FM; %)	35.4 (32.0−39.5)
Fat mass index (FMI; kg∙m^−2^)	11.8 (10.1−13.9)
Fat-free mass (FFM; %)	64.6 (60.5−67.5)
Peak oxygen uptake (VO_2peak_; mL∙kg^−1^∙min^−1^)	39.9 (36.1−41.7)
Maximum heart rate (HR_max_; beats∙min^−1^)	183 (175−187)
Maximum blood lactate (LAb; mmol∙L^−1^) ^a^	6.0 (5.0−7.0)
Maximal fat oxidation (MFO; mg∙kgFFM^−1^∙min^−1^)	4.1 (3.6−4.7)
VO_2_ (L∙min^−1^) *	1.3 (1.2−1.4)
*%* VO_2peak_ (mL∙kg^−1^∙min^−1^) *	35.9 (33.1−39.2)
HR (beats∙min^−1^) *	102 (96−107)
LAb (mM) *	0.9 (0.7−1.2)
Self-perceived exertion (1 to 10) *	1 (0−2)
Energy expenditure (*EE*; kcal∙min^−1^) *	6.6 (5.8−7.1)

* At maximal fat oxidation exercise intensity (FATmax).

**Table 2 ijerph-18-06945-t002:** Pearson’s product-moment correlations.

	Body Mass Index (kg∙m^−2)^	Fat Mass (g∙100 g^−1^)
MFO (mg∙kgFFM^−1^∙min^−1^)	0.49 (0.07, 0.76) *	0.64 (0.29, 0.84) **
FATmax (%VO_2peak_)	0.01 (−0.42, 0.44)	0.09 (−0.36, 0.50)
TFO (mg∙kgFFM∙min^−1^)	0.38 (−0.06, 0.70)	0.63 (0.27, 0.83) **
RER_peak_ (VCO_2_∙VO_2_^−1^)	−0.48 (−0.76, −0.06) *	−0.67 (−0.85, −0.33) **

Maximal fat oxidation (MFO), total fat oxidation (TFO), exercise intensity at which maximal fat oxidation occurs (FATmax), maximum respiratory exchange ratio at the onset of exercise (RER_peak_); * *p* < 0.05, ** *p* < 0.01.

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
