# Peer review of "Exercise Fat Oxidation Is Positively Associated with Body Fatness in Men with Obesity: Defying the Metabolic Flexibility Paradigm"

_ijerph, 2021, doi:10.3390/ijerph18136945_

Round 1

Reviewer 1 Report

The authors have attended to the comments suggested to improve the manuscript and should be accepted. I am satisfied with the response

Author Response

Thank you for the comments and suggestions

Reviewer 2 Report

None

Author Response

Thank you for your comments and suggestions

Reviewer 3 Report

Abstract

  • Page 1, line 17: Please reword to say “Obesity is thought to be associated with a reduced capacity…”
  • Page 1, line 28: Please reword to say “controlling for CRF”.
  • Page 1, line 25: R2 values can’t be negative.  Should this be r = -0.54?
  • Please make sure to include p-values with all Pearson r values and R2 There were some results mentioned in the literature for which p-values weren’t provided.  This needs to be consistent.  Without state p-values, the reader is not able to interpret the significance of the results, or agree with the concluding statement that MFO and RERpeak are valid indicators of TFO during steady-state exercise at FATmax.

Introduction

  • Please make sure that you are consistent when you state “FATmax.” In some places, it was written as “Fatmax.”  Please proofread your document and make corrections as needed (one example being on Page 1, line 63).
  • Page 1, line 41: Again, please reword to say “associated with obesity.”
  • Page 2, line 49: Please correct “perform” to “performed”.
  • Page 3, line 80: Please correct “metabolic flexible” to “metabolically flexible”.
  • Page 3, lines 90-91: Please reword this sentence to say, “…are associated and truly represent…”
  • Page 3, lines 99-100: This final sentence of the Introduction does not entirely make sense.  It is not clear what you mean by “the last.”  Please reword to make it a complete and clear sentence. 

Methods

  • Page 4, line 124: I am assuming that all 3 laboratory sessions performed between 8:00-11:00 h and after 10-12 h of overnight fasting.  This is not entirely clear from the text.  Please say if these conditions were applied to all laboratory testing sessions or only the first evaluation session.
  • Page 5, line 183: Please change the last word on this line from “de “ to “the”.
  • Page 5, line 200: Please change “realized” to “performed”.
  • Page 5, line 201: Please change “data was reported” to “data were reported”.

Results and Discussion

  • Page 6, lines 206-207: These sentences need to be reworded some.  In particular, it is fine to say that “all participants were considered obese”.  Also, it is better to say that the participants’ cardiorespiratory fitness was low, rather than saying that they were physically unfit.  Since ACSM publishes VO2peak norms for sex and age, please reference these ACSM norms when stating that participant cardiorespiratory fitness was low.
  • Page 6, Table 1: Please say “body mass” rather than “body weight”.
  • Page 7, lines 225-233 and Page 7, figures 4A-D: Should there be inferential statistics to accompany this substrate utilization data analysis?  It is difficult to interpret whether these changes are significant or not.
  • Page 9, line 251: The beginning of this sentence is awkward.  Do you mean to start with “In spite of the fat oxidation increase…”?
  • Page 9, line 258: Please change “propose” to “proposed”.
  • Page 9, line 270: Instead of saying “The here reported results…”, please reword to say, “The results of the present study agree with…”
  • Page 9, line 270: This should say, “class III obesity”.
  • Page 9, line 275: Please change “acids” to “acid”.
  • Page 9, line 276: Please place the word “and” before the Roman numeral II.  Also, please change “triglycerides” to “triglyceride”.
  • Page 9, line 277: Please change “mechanism” to “mechanisms”.
  • Page 9, line 279: Again, please change “the here observed results” to “the results of the present study”.  Also change “differs” to “differ”.
  • Page 9, lines 281-282: This sentence needs to be reworded to make it a complete sentence. Also, “participant’s” should be changed to “participants’ “ (e.g. the apostrophe should be placed after the word).
  • Page 10, lines 295-304, 312-314: Please be aware that the word “Metinflex” starts to appear, whereas it has been called “Metflex” in other parts of the manuscript.  Please revise in order to be consistent throughout the manuscript.
  • Page 10, line 315: Please change “would lead” to “may lead”.
  • Page 10, line 323: Please change “raised’ to “rose”.
  • Page 10, line 324: Please change “increases” to increased”.
  • Page 10, line 325: Please change “reaches” to “reaching”.
  • Page 10, line 330: Please change “a higher individuals CRF” to “higher participant CRF”.
  • Page 10, line 331: Please change the word “works” to “investigations”.
  • Page 10, lines 334-335: Please change this sentence to say, “because metabolic rate augments several fold above resting values.”
  • Page 10, lines 337-339: Please check the grammatical correctness of this sentence…it is a bit awkward to read.
  • Page 11, line 346: Please reword to say, “the first minutes after exercise onset,”

Author Response

This manuscript is a resubmission of an earlier submission. The following is a list of the peer review reports and author responses from that submission.

Round 1

Reviewer 1 Report

Thank you for the opportunity to review this important and well-written manuscript. Although, my comment is not much.

In line 162: Does instrumental jazz actually favor relaxation? Could instrumental jazz not be an influencer or stimulator? I am concerned about this. 

In line 167: I want to suggest that "Before" should be written, such as "Prior to the test performed on a Quinton treadmill" for clarity. 

Reviewer 2 Report

Introduction

Overall, the introduction can be shortened.

Line 38-39: Sentence beginning with “In this aspect” can probably be removed. I am not familiar with the several “markers” of metabolic flexibility mentioned, so maybe these “markers” should be identified? Metabolic flexibility in and of itself is the ability to switch substrate use to substrate availability.

Line 42-43: Metabolic inflexibility is/can be attributed to more than reduced muscle CPT-1 and CSA.

A major factor also regarding measurement of metabolic flexibility is diet intake which is not discussed in the introduction at all. Overnutrition (overfeeding) can affect metabolic flexibility as well as the macronutrient consumption prior to the measurement of metabolic flexibility. This should be addressed.  

Methods

What exactly are the authors defining as metabolic flexibility “markers?” It is the reviewer’s perception that metabolic flexibility is defined as the difference in substrate utilization between two conditions (for example: fasted/fed or fasted/insulin-stimulated) and can the body (quickly) adapt to what is being provided (ΔRER).

This study seems more like a design of capacity for fat oxidation in obese men, and not a test of metabolic flexibility. I am confused with the long focus of metabolic flexibility within the introduction, and then it is not addressed or calculated within the methods. It is possible the authors could take the difference in RER from rest to the RER measured at the intensity of identified MFO and relate that to cardiorespiratory fitness? So the ΔRER (RERrest – RERMFO) would indicate the person’s ability to utilize fat as a substrate during exercise at an intensity where fat should be the primary substrate.

Results

There is not a discussion of how metabolic flexibility is reflected in these results.

Line 214: “participants were obese” should be rephrased to reflect people-first language.

Discussion

The discussion is lengthy and could be shortened.

Reviewer 3 Report

Abstract

  1. Page 1, line 17: Please correct “liked” to “linked.”
  2. Page 1, line 18: Please correct “no” to “not.”
  3. Page 1, line 22: Please correct “on” to “in.”
  4. Page 1, line 23: Please correct “after” to “afterward.”
  5. General comment: Several abbreviations are used throughout the abstract; however, their meaning is not defined or explained.  This makes it difficult for the reader to fully understand the variables that were measured in this investigation, and thus the full meaning of the results.  Please address and clarify this.

Introduction

  1. Page 2, line 53: Please correct “They said” to “The authors commented that”.
  2. Page 2, line 58: Please correct “Respect to” to “With respect to.”
  3. Page 2, lines 67-68: This is not a complete sentence as it is currently written.  Please reword.
  4. Although you cite the sources for Figure 1 in the body of the text, please make sure that you have obtained the necessary copyright allowances to reprint these figures in your manuscript.
  5. Page 3, lines 89-103: This section needs to be reworded better in order to lead the reader from the background portion of the Introduction to the purpose statement, rationale for the study, and authors’ hypothesis/hypothesis.  For example:
  6. Please better explain why latency and fat oxidation increase are not reliable indicators of Metflex and why the RERpeak at exercise onset would be a better indicator of Metflex. Is this your hypothesis statement or is this based on previous research (yours or someone else’s)?
  7. Page 3, line 95: Please provide specific examples/rationale regarding what you mean by “increasing biases in the results interpretation and heterogeneity across studies.”
  8. Please provide a hypothesis statement after your Purpose statement.
  9. Page 3, line 103: I would recommend omitting “keeping VO2peak value as a constant” from this sentence.  This can be described in the Methods.

Methods

  1. Page 3, line 106: Please correct “obese males” to “males with obesity.”  This correction emphasizes to the readers that the participants are people first, and are not defined by their condition.  Please make this correction where is appears in other places in the manuscript.
  2. Page 3, lines 108-112: Please describe your study’s inclusion criteria in text form rather than bulleted form. This can appear within the first paragraph of your Methods section, starting on line 107 right after the first sentence that describes your participants.
  3. Page 3, line 113: Instead of saying “for the latter,” I recommend that you use more specific language such as “Participants reported their physical activity habits by recording…”
  4. Page 3, line 112: Please provide a short justification for why you operationally defined < 60 minutes as the cut-point for not being engaged in regular physical activity.
  5. Page 4, lines 124-125. Please reword this sentence to say “Once recruited, participants reported to the Exercise Physiology Laboratory on three separate occasions.”
  6. Page 4, line 126: Please correct “measure” to “measured.”
  7. Page 4, line 130: Please correct “exercise tests were” to “exercise test was.”
  8. Page 4, line 138: Please correct “remembered” to either “recalled” or “reported.
  9. Page 5, line 205: Please correct “controlling the variables by VO2peak” to “controlling for VO2peak.”

Results

  1. Please also include the physical characteristics of the participants, either within Table 1 or within the text at the beginning of the Results section. Specifically, please include age, race/ethnicity, height, and body mass.
  2. Page 6, line 215: I agree that the mean VO2peak of the participants in this study is on the low side.  However, please clarify what reference you are comparing the participants’ VO2peak to.  For example, are you comparing this mean VO2peak value to a set of normative data?  Please provide the reference.
  3. Page 6, line 216: Please clarify what you mean by “fat oxidation raised until 4.14 +/1.25 mg/kgFFM/min.” Do you mean that fat oxidation started at one value and then increased to this specified value?  Please address this.
  4. Table 1: Please correct the units for blood lactate. It should be milimoles per liter (mmol/L).

Discussion

  1. Page 9, line 255: Please specify “other health problems.”
  2. Page 9, line 257: I would recommend rewording “testing its association with body fatness by keeping VO2peak value as a constant” with “testing its association with body fatness while controlling for VO2peak.”
  3. Page 10, lines 259-264: Please explain why the results of your study are in contrast to those of previous studies. Are there key differences between your study and previous studies to suggest why the outcomes differ?  Please address.
  4. Page 10, lines 264-265: I advise that you soften your language somewhat.  Instead of saying “confirm,” I advise saying “suggest.”  Similarly, I would advise replacing “truthful” with “valid”. Since your study is one in the literature and has a fairly small sample size, more studies are warranted to truly confirm your study’s results.
  5. Page 10, lines 268 and 274: Again, you are getting ahead of yourself when subsequent studies can rely on the results of your study to use MFO and RERpeak to calculate Medflex grade and its associations with metabolic syndrome.  Again, please soften your language to say that the results of your study may potentially form the basis for using MFO and RERpeak in these ways as you state, and that future studies may want to consider this approach.
  6. Page 10, lines 288-299: Again, please explain why the results of your study contrast those of previous studies.  Are there key differences between your study and previous studies to suggest why the outcomes differ?  Please address.
  7. Page 11, line 308: Please correct “on” to “in.”
  8. Page 11, line 349-350: Please provide a reference for this sentence.  Also, muscle contractions do not directly elicit catecholamine release from the adrenal medulla.  Nor do they directly elicit calcium release from the sarcoplasmic reticulum (it is the calcium release from the SR that facilitates the formation of crossbridges in order to generate muscle force and power).  Please also correct this sentence so that the information is physiologically accurate.
  9. Page 12, line 358: In addition to inhibiting CPT-1, decreased muscle pH also decreases the activity of citrate synthase (major enzyme of the Krebs Cycle).
  10. Page 12, line 360: The word “internalization” is confusing here. Please substitute with a different word or phrase.
  11. Page 12 line 362: Please specific the two kinases to which you refer.
  12. Page 12, line 363: Please substitute “epinephrine and norepinephrine” for “adrenaline.”
  13. Page 12, lines 374-381: Please also include in this section, a reiteration of the novel aspects of your study, and what novel information it adds to the literature.
  14. Page 12, line 386: Please correct “obese male individuals” to “male individuals with obesity.”
  15. Page 12, line 387: Please describe other populations in whom this research is recommended to occur (e.g. females, individuals with specified health conditions, individuals who are younger adults or older adults, etc.).
  16. Page 12, lines 383-393: Please also comment on whether your study sample size is a limitation.  How does it compare to other studies in the literature?  If it is similar or larger than previous studies, then you may consider your sample size to be a potential strength. However, if it is smaller than previous studies, then your sample size is a limitation and should be addressed and explained as such.
  17. Page 12, lines 395-402: Please also describe the practical or clinical implications of your study’s findings.